# Robust and Multi-Objective Pareto Design of a Solenoid

Krisztián Gadó [ID] and Tamás Orosz *[ID]

Montana Knowledge Management Ltd., 1111 Budapest, Hungary; gado.krisztian@montana.hu
* Correspondence: orosz.tamas@montana.hu

**Abstract:** The optimization of the design of a practical electromagnetic device involves many challenging tasks for new algorithms, especially those involving numerical modeling codes in which objective function calls must be minimized for practical design processes. The Compumag Society provides openly accessible, challenging benchmark problems (TEAM problems) for testing novel numerical solvers. This paper deals with a novel solution for the multi-objective TEAM benchmark problem. This solenoid design test problem aims to search for the optimal shape of a coil, which ensures a uniform field distribution in the control region, while the sensitivity and the mass/DC loss of the coil are also considered in the context of robust design. The main differences from the previously published solutions are that the proposed methodology optimizes all three objectives together, not only as two independent two-dimensional sub-problems. We considered the asymmetrical cases in the solution and found that the symmetrical solutions always produced better uniformity and sensitivity measures. However, the difference between the symmetrical and asymmetrical solutions is insignificant for these objectives. Despite the fact that the cheapest solutions are symmetrical setups, they perform worse than the cheapest asymmetric ones in these uniformity and sensitivity criteria. Therefore, some asymmetric solutions that were previously neglected from the solution space can be competitive and interesting for practical design.

**Keywords:** optimization; electrical machines; design optimization; finite element method

## 1. Introduction

The practical design of an electrical device usually leads to a multi-objective optimization task. These problems must involve the resolution of many computationally expensive finite-element-methodology-based numerical field calculations. The goal of the approaches that are applied is to reduce the number of function evaluations or to reduce a numerical model's computational cost without a significant loss of accuracy [1,2]. During industrial design processes, not only the physical parameters of a machine, but also the manufacturing tolerances and the different uncertainties should be considered right from the beginning of the design process [3–6].

The goal of a design optimization task differs when it is assessed from a mathematical or industrial perspective. Mathematically, the goal of a design optimization task is to find not only a better solution, but also the global optimum of the task, if it exists [7]. However, models that can be used for optimization are usually simplified ones that do not consider many factors, which is important for the easy and robust manufacturability of the product. Moreover, the design of an electrical device usually leads to a general nonlinear optimization problem. The result of these optimization problems is a Pareto-front, i.e., a set of non-dominated solutions [2,8,9]. From the industrial point of view, solutions that are better than the previous ones are usually considered as optimal ones because many factors are neglected during industrial optimization processes.

The TEAM (Testing Electromagnetic Analysis Methods) (https://www.compumag.org/wp/team/, (accessed on 1 June 2021)) benchmark problems offer a wide variety of test problems for benchmarking the accuracy of numerical solvers, and they are openly available from the website of the International Compumag Society [10]. The subject of

our analysis is a multi-objective Pareto optimization of a solenoid, which is cataloged as the 35th benchmark problem in the list [11,12]. The goal of this benchmark is to design a coil that generates a uniform magnetic field in a given control region (Figure 1). The sensitivity to positioning errors of the turns and the power loss or mass of the given design are also considered. This seemingly simple test problem is inspired by a bio-electromagnetic application for Magnetic Fluid Hyperthermia (MFH), where a uniform magnetic field is used to compare the magnetic properties of different nanofluids [5,13–15]. A similar design task should be solved during the design and optimization of induction heating or induction brazing processes [16–22]. The solution of this test problem requires the resolution of a three-objective optimization problem in which the objective functions depend on finite-element-method-based calculations, and one of the objective functions considers the robustness of the solution during the optimization process.

Many solutions were proposed to resolve this problem in the literature [23]. The first paper proposed the DC problem and computed this optimization task through a gradient-based evolutionary algorithm (EA) search [12]. The following paper resolved the same problem with different EAs [24]. This comparison has great importance because several EAs were used to determine inverse electromagnetic tasks. Due to Wolpert's "No-free-lunch" theorem [2,25,26], these metaheuristics must be benchmarked with each other for every kind of optimization task in order to select the most appropriate one. Seo et al. [27] solved this problem with a design sensitivity analysis, which provided almost the same optimization result as the gradient-search-based evolutionary algorithms in a much shorter time. These authors used FEM solvers to calculate the magnetic field. Karban et al. [5] proposed a semi-analytical formula and validated its precision with an hp-adaptive FEM solver [28]. The goal of this semi-analytical formula was to accelerate the solution speed of the magnetostatic problem. This problem was solved by other authors with different evolutionary and genetic algorithms, such as Non-Dominated Sorted Genetic Algorithm (NSGA-II) [29], Wind-Driven Optimization [30], the Micro-Biogeography Inspired Multi-Objective Optimization ($\mu$-BiMo) [31], and the Migration Non-Dominated Sorted Genetic Algorithm (MNSGA-III) [32]. Another paper modified the excitation and solved this problem with the time-harmonic regime as a 2D or a 3D problem, considering the proximity effect of the windings [23]. However, these previously published solutions did not consider the measurement limits and other confounding factors, such as the Earth's magnetic field, which can be greater than the contributions of these effects or the difference between two designs.

This paper proposes an approach for the original DC problem that is different from those of the papers that were mentioned earlier [12,24,27,28], in which the original three-objective problem was resolved as two separate bi-objective problems. The proposed task is handled as a three-objective optimization task because this form of the problem fits better for real-life design tasks. The previous papers excluded asymmetrical solutions from the optimization. However, due to the non-linearity of the optimization problem, some asymmetrical solutions can be competitive with some symmetrical solutions. This 3D solution space is analyzed in this paper, and the analysis makes two other modifications to the parameter space of the problem. Firstly, the boundaries of the radii are changed. Secondly, the number of turns is varied, and the results of these three separate analyses are compared in the paper. The project files of the proposed analysis can be downloaded from the Artap project's homepage (https://github.com/artap-framework/artap/tree/master/examples, (accessed on 1 June 2021)).

## 2. Formulation of the Problem

The goal of this optimization is to create a uniform field distribution in the control zone with a solenoid [12,23,24]. The solenoid is composed of 20 series of connected, singular turns, with a radial position varying from 5 to 50 mm. These turns have exactly the same size. The width of each turn is $w = 1.5$ mm, the height is $h = 1.0$ mm, and the prescribed DC current density is $j_\phi = 2 \frac{A}{mm^2}$. The flux density should be $B(r,z) = (0, B_0)$ with $B_0 = 2$ mT in

the controlled region. The model of the optimized coil is shown in Figure 1, where the green rectangles show the controlled region, and the yellow ones represent the different turns. The main difference from the original description of the TEAM 35 benchmark problem is that the full coil is modeled for asymmetrical calculations, not just the upper half of the solenoid (the description of the examined TEAM benchmark problem can be found at https://www.compumag.org/wp/wp-content/uploads/2021/07/problem-35.pdf, (accessed on 1 September 2021)). The quality of the uniform magnetic field is assessed by using the point values of the magnetic field, which are evenly spaced among 100 points of the controlled region.

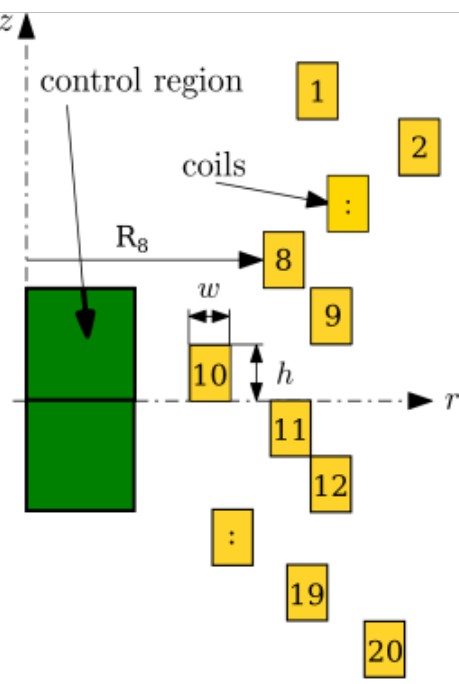

**Figure 1.** The examined geometry in an axisymmetric arrangement. The green area shows the controlled region in which the magnetic field is considered. Every single turn of the coil is denoted by yellow rectangles; their radii are optimized turn by turn.

Three different objective functions were used to measure the quality of the different solutions. The first one describes the uniformity of the magnetic field in the examined region; the $z$-component of the flux density is sampled in a $10 \times 10$ grid. The function takes the maximal difference from the intended flux density ($B_{0z} = 2$ mT) into consideration:

$$F_1 = \max |B_{0z} - B_{iz}|, i = 0, \dots, 99. \tag{1}$$

The second function considers the robustness of a given solution. Let $\mathbf{B}(\mathbf{R})$ be the flux density values (2 rows, 100 columns) at a given input $\mathbf{R}$. After that, the $\mathbf{B}^+(\mathbf{X} + \Delta\xi)$ and $\mathbf{B}^-(\mathbf{X} - \Delta\xi)$ vectors need to be computed, where $\Delta\xi = \pm 0.5$ mm and $\mathbf{B}^-(\mathbf{X} - \Delta\xi)$ represents the magnetic flux density change in the case of a 0.5 mm positioning error in a turn. $F_2$ can be defined in the following way:

$$F_2 = \max\{\|\mathbf{B}_i^+ - \mathbf{B}_i\| + \|\mathbf{B}_i - \mathbf{B}_i^-\|, 0 \le i \le 99\}, \tag{2}$$

where $\|.\|$ means the Euclidean norm and $i$ represents the measurement point in the control region.

The third function represents the mass or the DC loss of the coils. These quantities are proportional to the input. The summation of $\mathbf{R}$ is given by $F_3$:

$$F_3 = \sum R_j. \tag{3}$$

where $R_j$ represents the radii of the separate turns. There are 20 different turns in the assembled coils, and their distance from the z-axis can be defined separately (Figure 1).

### 2.1. Modeling and Optimization Frameworks

The coil model described above was modeled with two different FEM tools: Agros2D, an hp-adaptive FEM solver, and a widely used tool, FEMM, which was used to solve this magnetostatic problem [33,34]. The model was defined by the Adze-modeler (Figure 2), which allowed us to switch between the applied solvers with one command and connect the realized model with Ārtap (Ārtap is available for download from: http://www.agros2d.org/artap/ (accessed on 1 June 2021) [35]). The workflow for the Adze-modeler can be seen in Figure 2. Different pieces of the geometry can be imported from different file types and can be exported to various FEM solvers by using the same description of the physical model. These parametric FEM models are generated by a function call from Ārtap, which is an optimization framework for robust design optimization. It was developed within the Department of Theoretical Electrical Engineering at the University of West Bohemia in conjunction with a fully hp-adaptive FEM-solver: Agros Suite or Agros2D [28,36,37]. It provides a simple, general interface for facilitating the solution of real-life engineering design problems. The code contains evolutionary and genetic algorithms, wrappers for derivative-free methods, machine learning methods, and an integrated FEM solver. The goal of the realized multi-layered architecture is to separate the problem's definition from those of optimization algorithms and other artificial-intelligence-based methods and to provide automatic parallelization and database connection for the applied algorithms [5,35].

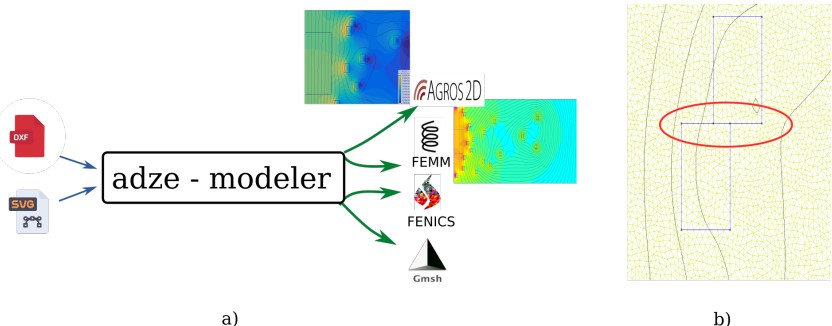

a)　　　　　　　　　　　　　　　　　　　　　　　b)

**Figure 2.** The image (**a**) shows the functionalities used and the workflow that was realized via the Adze-modeler. (**b**) The usage of the collision detection function, which automatically replaces the overlapping edges due to the optimization process.

### 2.2. Model Validation

The solenoid was simulated in two different ways in this paper. Firstly, the assumptions of symmetry were used, and only the upper half of the model was calculated with the FEM solvers. This model contains the first ten turns, and a Neuman boundary condition is applied at the $z = 0$ axis. This model is exactly the same as that used in the reference calculations [12]. However, the second model contains all 20 turns without using any assumptions of symmetry. The goal of this analysis is to let the optimizer select anti-symmetric solutions. These models were prescribed in Adze-modeler, which could export them for the FEMM or Agros2D models. The following test case was selected from Table 1 [12] to validate the correctness of our FEM models and the calculation of the objectives. Only the results of the symmetrical 10-turn model are presented in this paper, as we found the same results with the asymmetrical 20-turn solenoid model.

The following vector contains all of the design variables for the selected test case (Table 1, [12]):

$$X_1 = [6, 7, 8, 9, 10, 11, 12, 13, 14, 15]. \tag{4}$$

The simulations were performed with Agros2d and FEMM. In the case of Agros2d, an hp-adaptive mesh was set with 0.001% tolerance, while the mesh size was set to 0.5 mm in FEMM. The results and the settings used are summarized in Table 1. It can be seen that the resulting values are in a relatively good agreement with the reference calculations. The absolute value of the differences is a scale of magnitude smaller than the Earth's magnetic field, which can be a possible measurement limit in practice. The results of the $F_3$ objective calculations are not shown in Table 1 because the value of this function does not depend on the finite element models; it is simply calculated from the sum of the input vectors.

**Table 1.** The results of the validation run for the $X_1$ input compared to the reference values of $F_1$ and $F_2$.

|  | $F_1$ [T] | $F_2$ [T] | $\Delta F_1$ [T] | $\Delta F_1$ [%] | $\Delta F_2$ [T] | $\Delta F_2$ [%] |
|---|---|---|---|---|---|---|
| Reference | $8.18 \times 10^{-4}$ | $3.01 \times 10^{-4}$ | - | - | - | - |
| FEMM | $8.40 \times 10^{-4}$ | $2.91 \times 10^{-4}$ | $-2.20 \times 10^{-5}$ | 2.69% | $1.00 \times 10^{-5}$ | 2.20% |
| Agros2D | $8.36 \times 10^{-4}$ | $2.93 \times 10^{-4}$ | $-1.80 \times 10^{-5}$ | 2.20% | $8.00 \times 10^{-6}$ | −2.66% |

### 2.3. Sensitivity Analysis of the FEM Models

Another comparative analysis was made for calibration purposes. The analysis aimed to minimize the solution time and the computational demand of the optimization task by selecting the smallest mesh that was large enough to solve the task with the required accuracy. The required accuracy was 1% in the $F_1$ and $F_2$ metrics because it meant 20 μT in our problem, which is comparable with the Earth's magnetic field. Therefore, it is smaller than the precision of the measurement or the other neglected modeling details.

During the analysis, the Adze-modeler was used to convert and solve a randomly selected geometry for the different FEM tools. This model was solved with different mesh and solver settings (Table 2.) in FEMM [33] and Agros2D [28,36]. The sensitivities of the $F_1$ and $F_2$ objective functions with the different mesh settings were compared in a selected geometry (Figure 3).

**Table 2.** The settings applied for the FEMM- and Agros2d-based calculations.

| Parameter | FEMM | Agros2D |
|---|---|---|
| Problem type | | Magnetostatic |
| Analysis type | | Steady-state |
| Coordinate system | | axisymmetric |
| Polynomial order | 1 | 2 |
| Mesh settings | Smartmesh = Off | hp-adaptivity |
| Mesh size | 0.1–3 | tolerance = 5–0.005% |

In Agros2D, the polynomial order (p-adaptivity) was set to 2 for all cases. The mesh refinement (h-adaptivity) was considered with different error indicator settings from 5% to 0.005% (Table 2). FEMM can only use first-order polynomials and does not have any adaptivity. It has a "Smart Mesh" feature that is turned on by default. It generates a dense-enough mesh with Triangle [38] to ensure accurate calculations, but it cannot be parameterized. We set up the same mesh size in all regions by turning off this feature, and this mesh size changed during the comparison process. The settings applied are summarized in Table 2.

The $F_1$ and $F_2$ functions were calculated from the point values of the magnetic flux density of the control zone; hence, these functions can be sensitive to the numerical errors of the point values of the magnetic flux density calculations. It can be seen from Figure 3c,d that the $B_r$ and $B_z$ components are sensitive to the mesh applied at the top right corner of the control region (r = 5.0 mm, z = 5.0 mm). There is a huge difference in the

convergence speed of the two different FEM solvers applied. The smallest FEMM mesh contains about $5 \times 10^5$ nodes, whereas the Agros2D converges to the result in both $B_r$ and $B_z$. The difference in the case of the radial component is not significant. However, in the case of the axial component, the difference is significant (about 40%).

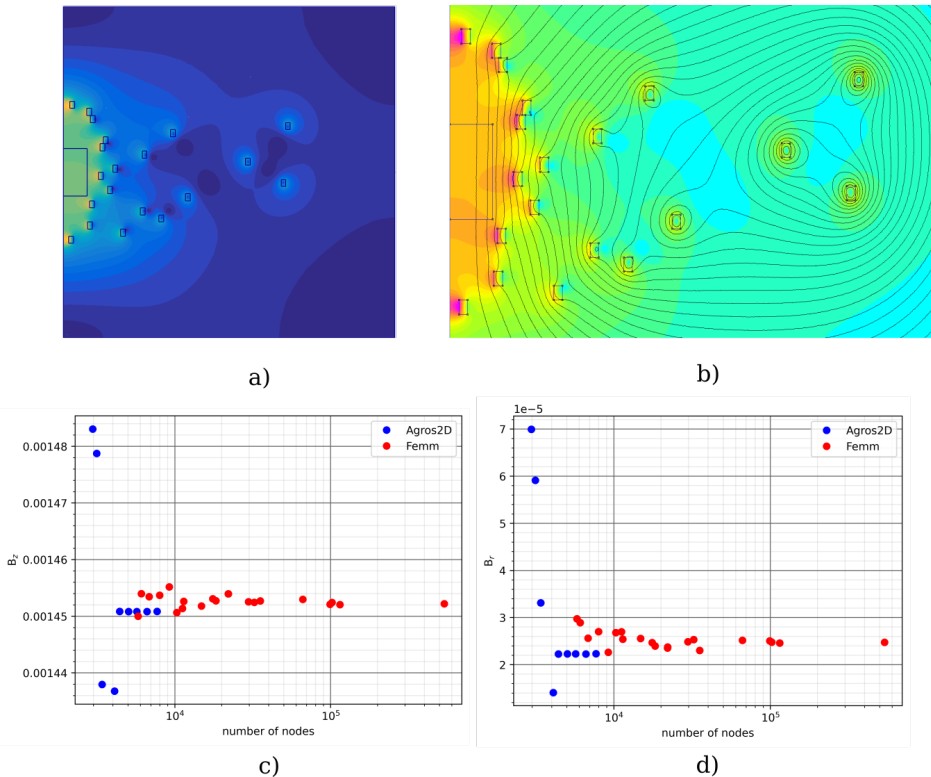

**Figure 3.** The pictures (**a**,**b**) show the geometry examined and the resulting flux distribution with the Agros2D (**a**) and FEMM software (**b**). The pictures (**c**,**d**) show the convergence of the radial and the axial component of the flux density in the selected ($r = 5$ mm, $z = 5$ mm) point.

The same solution is plotted in Figure 4 to visualize the differences in the solutions with the following settings: the error indicator was set to 1% in Agros2d and the smartmesh function was used in FEMM (Figure 4). There is a significant difference in the point values of the magnetic flux density on the right side of the examined region ($r = 5$ mm).

Figure 5 shows that these calculation errors have an effect on the objectives. $F_1$ considers the maximal difference from the expected value at a single point. These points can cause significant errors during the optimization. The sensitivity of the $F_1$ and $F_2$ functions to the mesh selection was examined. It is plotted in Figure 5, where the picture (a) justifies the above-mentioned assumption that the mesh selection has a significant (50%) effect on the values of $F_1$.

The $F_3$ function is independent of the mesh selection, and it has a local minimum when all of the radii have the minimum value because it simply depends on the geometry of the coil. Agros2D was used for further calculations with the following settings (Table 3) because it clearly outperformed FEMM during the analyses. The error indicator was set to 1%. Using a more precise calculation seemed pointless because 1% of our target value ($B_0 = 2$ mT) was only 20 μT, which is smaller than the Earth's own magnetic flux density, which would affect the calculation results. If the final application needs more precise results, this effect should also be considered with magnetic and other geometrical simplifications.

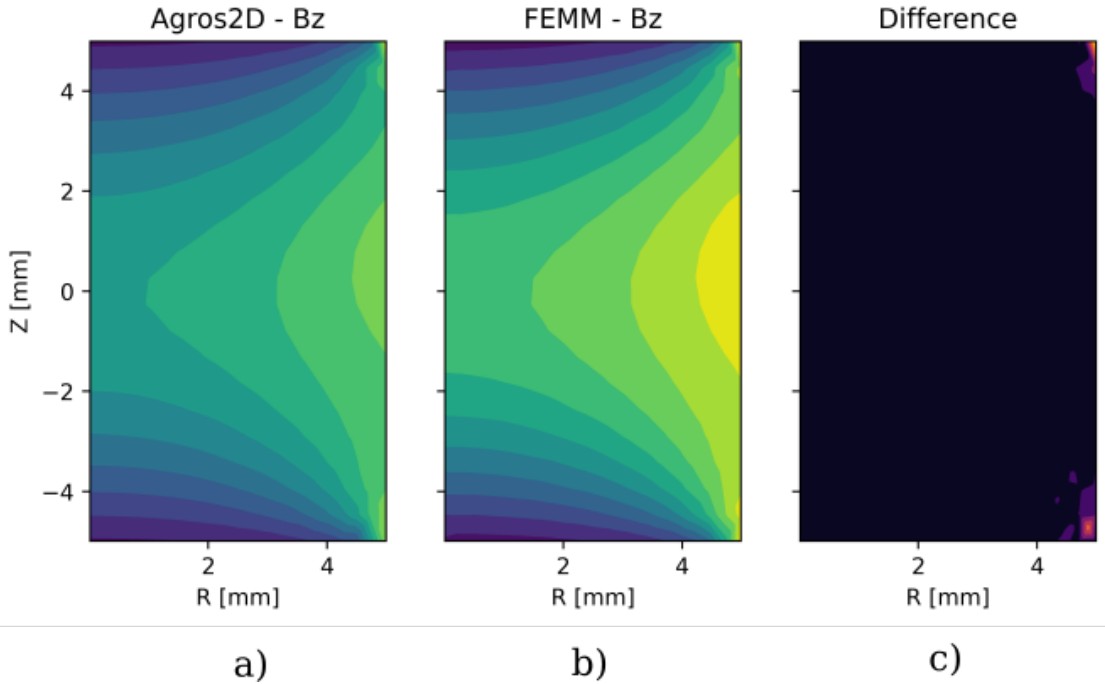

**Figure 4.** The pictures (**a**,**b**) show the calculated flux density values in the upper half of the examined region with the two types of software compared: Agros2d (**a**) and FEMM (**b**). The third picture (**c**) shows the dependence of the $B_z$ values on the meshing properties.

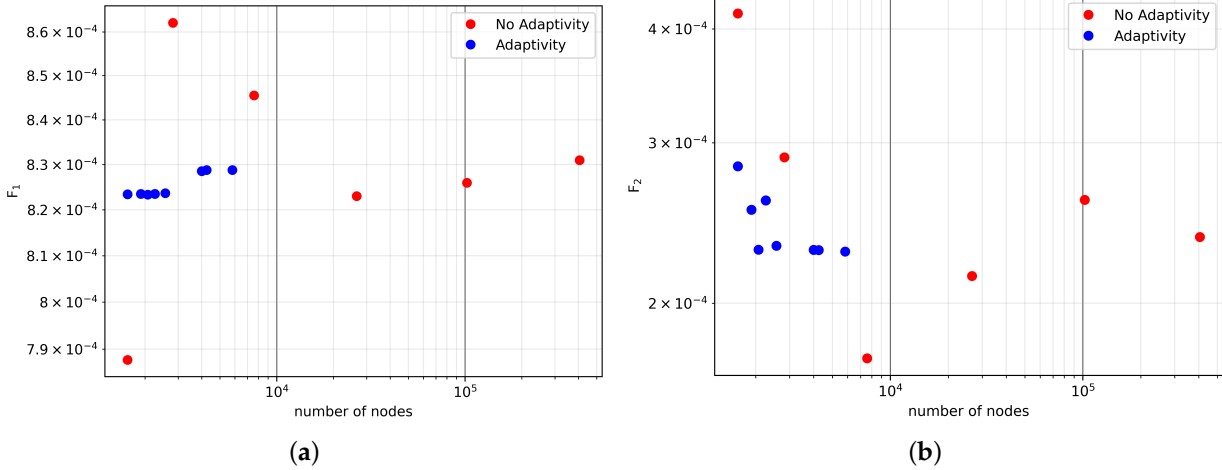

(**a**)　　　　　　　　　　　　　　　　　　　　　　　(**b**)

**Figure 5.** The convergence of the objective functions ($F_1$ and $F_2$) in terms of the number of nodes with and without hp-adaptivity. (**a**) The image plots the dependence of $F_1$, while (**b**) plots the dependence of the $F_2$ function on the number of nodes.

**Table 3.** The settings used for Agros2D during the optimization.

| Parameter | Agros2D |
| --- | --- |
| Problem type | Magnetostatic |
| Analysis type | Steady-state |
| Coordinate system | axisymmetric |
| Polynomial order | 2 |
| Mesh settings | hp-adaptivity |
| Mesh size (tolerance) | 1% |

## 3. Results and Discussion

### 3.1. Optimization of the Three-Objective Problem

The 35th TEAM benchmark problem was optimized as a three-objective optimization task. This approach is the first difference from the previously proposed solutions, where the whole problem was divided into two multi-objective optimization tasks [12,23]. The 2D segments of the three dimensional Pareto surface are plotted and examined in the following subsections and figures. All three objectives were considered during the optimization process because during the design of a product, all of these aspects should considered together. The symmetrical and asymmetrical solutions were optimized separately. The "symmetrical" model refers to the 10-turn geometry, where only the upper half of the solenoid is calculated during the optimization, while the asymmetrical 20-turn model makes it possible to optimize all of the turns separately. Both the "symmetrical" and "asymmetrical" models were calculated with Agros2D with the setup discussed above (Table 3).

The optimization was performed with Artap while using the NSGAII algorithm [29]. In all of the cases, the maximum number of generations was 250 with 100 individuals. In the symmetrical cases, the optimization contained 10 independent variables, while in the asymmetrical cases, the problem contained 20 independent variables.

The following analyses were made for the three cases with the three different settings:

(a)　The radii of the first four and the last four turns varied from 1 to 50 mm, while the radii could be changed from 5.5 to 50 mm in the case of the other turns.

(b)　The radii of all of the turns could be changed from 5.5 to 50 mm.

(c)　The number of turns was reduced to 12, and all of the radii could be changed from 5.5 to 50 mm.

The results of these different optimization tasks are discussed in the following subsections.

### 3.2. Optimization of Case (a)

First of all, the three-dimensional Pareto surface after the optimization is plotted in Figure 6. It can be seen that the shape of this function is very spiky, and it is hard to localize one distinct optimum. There are many local optima that are close to each other, but most of them are very sensitive to the parameter changes. Optimizing the given solenoid for only one of the selected goal functions can easily lead to a non-robust solution.

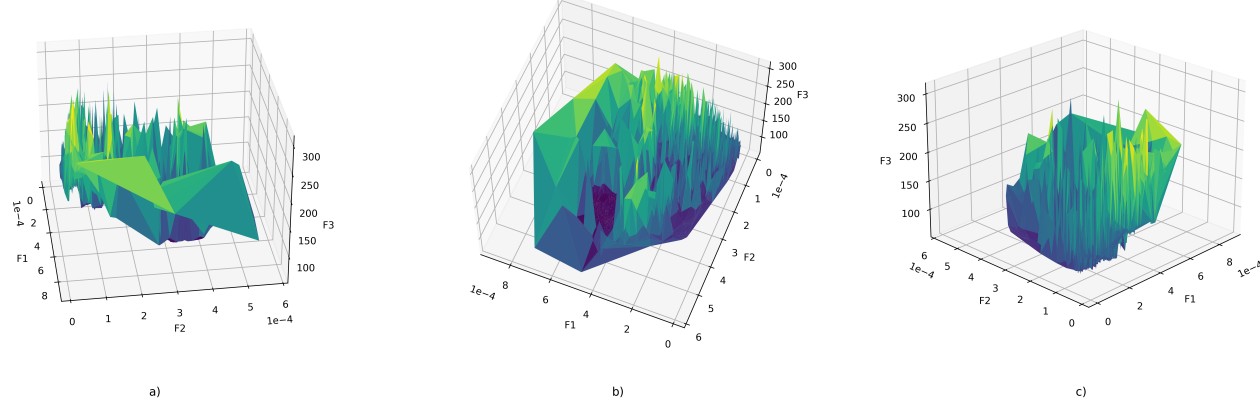

**Figure 6.** The plots (**a–c**) depict the shape of the optimized $F_1$, $F_2$, and $F_3$ objectives, (**a–c**) images plots the 3D surface of the objective function from different views.

After the optimization was performed, the results were sorted based on the values of the different objective functions, as can be seen in Figures 7–9. The different sortings indicate different priorities, but the optimization was performed with all three functions considered. For example, the $F_1$ sorting means that the last generation of solutions was sorted based on the $F_1$ value, and then the first 100 were selected. This sorting aims to

show that the asymmetric solutions can be used just as well as the symmetric ones, and for some criteria, they even outperform the symmetric solutions.

Initially, the solutions from the three-dimensional Pareto plot that had the best values for $F_1$ were examined (Table 4). This means that these solutions have the best performance for the uniformity objective. As shown in Figure 7a, the $F_1$ values are in the range of 10–40 μT, which is within the measurement's error. This means that they can be considered equal. This holds on the $F_2$ axis as well. In terms of price, both variant groupings are in the same price range, but the symmetric setups have a slight advantage. Therefore, the best solutions are symmetrical ones, but there is no significant difference between the best symmetrical and asymmetrical solutions if we consider the uniformity of the solutions.

**Table 4.** Solutions based on the $F_1$ sorting in case (a).

| | $F_1$ | $F_2$ | $F_3$ | $R_1$ | $R_2$ | $R_3$ | $R_4$ | $R_5$ | $R_6$ | $R_7$ | $R_8$ |
|---|---|---|---|---|---|---|---|---|---|---|---|
| Symmetric | $1.02 \times 10^{-5}$ | $5.60 \times 10^{-5}$ | $1.18 \times 10^2$ | 3.55 | 6.09 | 1.29 | 5.16 | 5.76 | 7.22 | 7.00 | 7.50 |
| Asymmetric | $2.07 \times 10^{-5}$ | $6.16 \times 10^{-5}$ | $1.00 \times 10^2$ | 1.29 | 1.47 | 2.43 | 5.02 | 6.42 | 5.43 | 6.54 | 6.80 |

| | $R_9$ | $R_{10}$ | $R_{11}$ | $R_{12}$ | $R_{13}$ | $R_{14}$ | $R_{15}$ | $R_{16}$ | $R_{17}$ | $R_{18}$ | $R_{19}$ | $R_{20}$ |
|---|---|---|---|---|---|---|---|---|---|---|---|---|
| Symmetric | 7.50 | 7.90 | 7.90 | 7.50 | 7.50 | 7.00 | 7.22 | 5.76 | 5.16 | 1.29 | 6.09 | 3.55 |
| Asymmetric | 6.99 | 7.14 | 7.18 | 6.92 | 6.69 | 6.67 | 5.82 | 5.55 | 4.47 | 1.47 | 4.61 | 1.06 |

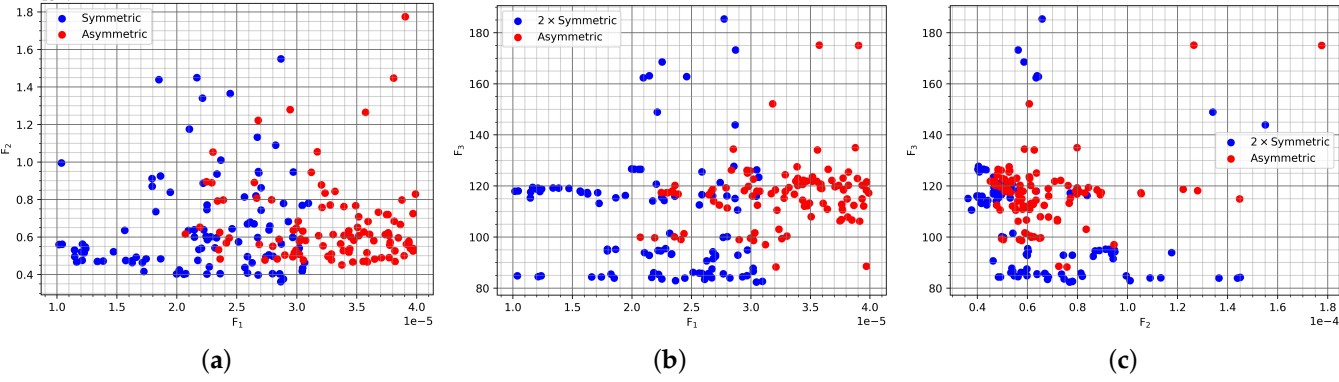

(a)  (b)  (c)

**Figure 7.** The image shows the first 100 individuals when they are sorted based on uniformity ($F_1$) in case (**a**). The blue dots show the symmetric setups, while the asymmetric solutions are depicted in red. (**a**) The distribution of individuals on the $F_1$–$F_2$ axis. (**b**) The distribution on the $F_1$–$F_3$ axis. (**c**) The distribution on the $F_2$–$F_3$ axis.

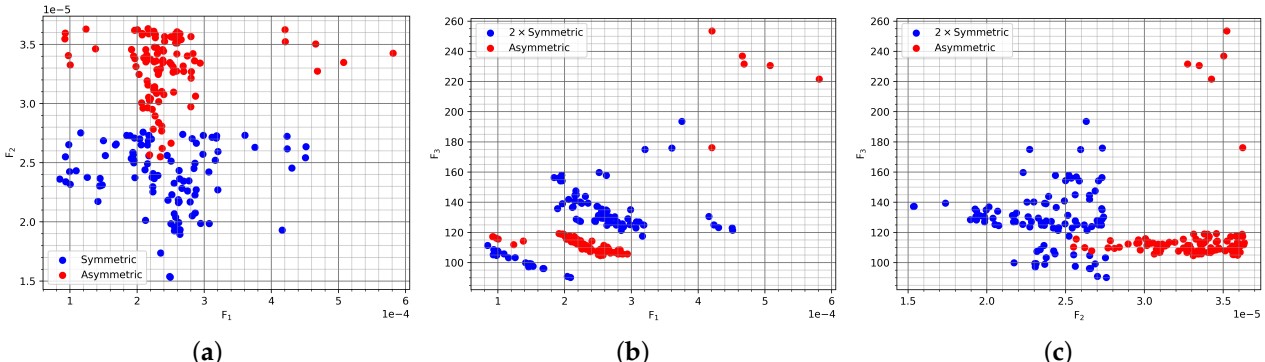

(a)  (b)  (c)

**Figure 8.** The first 100 individuals from the three-dimensional Pareto front were sorted based on their robustness ($F_2$) in case (**a**). The blue dots show the symmetric setups, while the asymmetric solutions are depicted in red. (**a**) The distribution of individuals on the $F_1$–$F_2$ axis. (**b**) The distribution on the $F_1$–$F_3$ axis. (**c**) The distribution on the $F_2$–$F_3$ axis.

Secondly, the most robust solution was sought, and it was a result of an $F_2$ sorting (Table 5). As shown in Figure 8a, the symmetric setups have an advantage, but they are in the region of the measurement error again, so they can be considered equally robust. In terms of uniformity, the difference is negligible. Regarding the price criteria, the symmetric setups can be slightly cheaper.

**Table 5.** Solutions based on the $F_2$ sorting in case (a).

| | $F_1$ | $F_2$ | $F_3$ | $R_1$ | $R_2$ | $R_3$ | $R_4$ | $R_5$ | $R_6$ | $R_7$ | $R_8$ |
|---|---|---|---|---|---|---|---|---|---|---|---|
| Symmetric | $2.49 \times 10^{-4}$ | $1.53 \times 10^{-5}$ | $1.37 \times 10^2$ | 6.82 | 6.42 | 3.55 | 1.97 | 5.33 | 7.07 | 8.99 | 8.69 |
| Asymmetric | $2.34 \times 10^{-4}$ | $2.55 \times 10^{-5}$ | $1.10 \times 10^2$ | 2.51 | 3.33 | 2.86 | 4.44 | 4.98 | 6.44 | 6.92 | 8.97 |

| | $R_9$ | $R_{10}$ | $R_{11}$ | $R_{12}$ | $R_{13}$ | $R_{14}$ | $R_{15}$ | $R_{16}$ | $R_{17}$ | $R_{18}$ | $R_{19}$ | $R_{20}$ |
|---|---|---|---|---|---|---|---|---|---|---|---|---|
| Symmetric | 9.76 | 9.99 | 9.99 | 9.76 | 8.69 | 8.99 | 7.07 | 5.33 | 1.97 | 3.55 | 6.42 | 6.82 |
| Asymmetric | 7.42 | 10.82 | 8.03 | 8.18 | 8.39 | 7.11 | 6.00 | 5.01 | 3.75 | 1.50 | 2.57 | 1.09 |

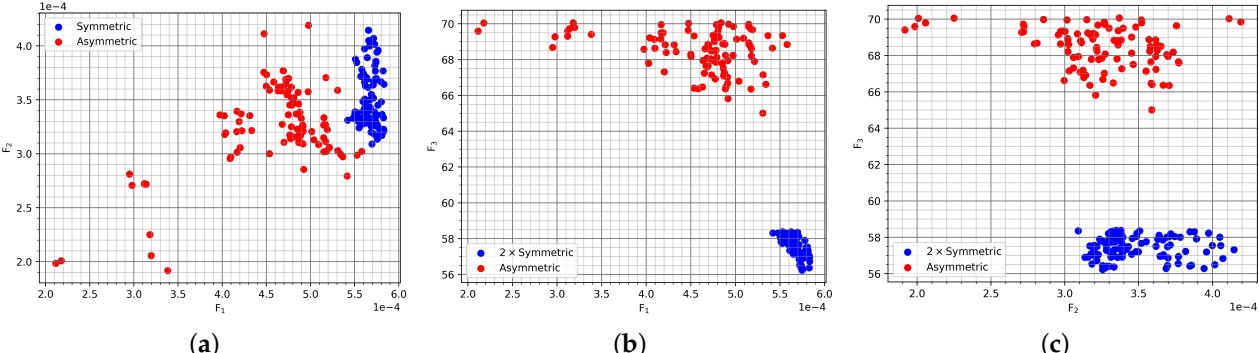

(a)  (b)  (c)

**Figure 9.** The image shows the distribution of the 100 cheapest individuals ($F_3$) in case (**a**). The blue dots show the symmetric setups, while the asymmetric solutions are depicted in red. (**a**) The distribution of individuals on the $F_1$–$F_2$ axis. (**b**) The distribution on the $F_1$–$F_3$ axis. (**c**) The distribution on the $F_2$–$F_3$ axis.

The goal of the third examination was to sort the results by their price ($F_3$) (Table 6). It can be seen in Figure 9 that the symmetric setups are cheaper by 12–20%, but choosing one of them would be a sub-optimal solution, since the asymmetric setups perform significantly better in terms of accuracy and robustness. The radii of the Pareto-optimal results are plotted in Figure 10a. This violin plot shows the approximate shape of the symmetrical and asymmetrical solutions. The optimizer can set tiny values of the radii for the coils that are placed above or below the homogenized region in these solutions. Most of the Pareto-optimal solutions have at least one turn that has a radius $\leq 5.5$. There is no big difference between the distributions of the radii for the symmetrical and asymmetrical cases. We can conclude that some asymmetrical results that can be competitive with the symmetrical ones exist.

**Table 6.** Solutions based on the $F_3$ sorting in case (a).

| | $F_1$ | $F_2$ | $F_3$ | $R_1$ | $R_2$ | $R_3$ | $R_4$ | $R_5$ | $R_6$ | $R_7$ | $R_8$ |
|---|---|---|---|---|---|---|---|---|---|---|---|
| Symmetric | $5.75 \times 10^{-4}$ | $3.26 \times 10^{-4}$ | $5.62 \times 10^1$ | 1.01 | 1.01 | 1.00 | 1.01 | 1.00 | 1.02 | 5.51 | 5.50 |
| Asymmetric | $5.31 \times 10^{-4}$ | $3.59 \times 10^{-4}$ | $6.50 \times 10^1$ | 1.14 | 1.12 | 1.01 | 1.29 | 1.03 | 5.48 | 5.65 | 5.75 |

| | $R_9$ | $R_{10}$ | $R_{11}$ | $R_{12}$ | $R_{13}$ | $R_{14}$ | $R_{15}$ | $R_{16}$ | $R_{17}$ | $R_{18}$ | $R_{19}$ | $R_{20}$ |
|---|---|---|---|---|---|---|---|---|---|---|---|---|
| Symmetric | 5.55 | 5.51 | 5.51 | 5.55 | 5.50 | 5.51 | 1.02 | 1.00 | 1.01 | 1.00 | 1.01 | 1.01 |
| Asymmetric | 6.03 | 6.62 | 5.69 | 5.67 | 5.73 | 5.54 | 1.31 | 1.37 | 1.17 | 1.07 | 1.08 | 1.26 |

If all of the objective functions are considered, then the symmetrical and asymmetrical variants perform similarly (Table 7) because the difference between them lies in the region of tolerance. The fact that these solutions vary in the values of their radii implies that more than one solution exists for this problem. In Figure 10a, one can see the distribution of the values of the radii for the last 100 individuals with different setups. In Figure 11a, the symmetric and asymmetric setups are compared. In this optimization, all coils are free to move within their logical boundaries. Both setups converge roughly to the same range. The first and last four coils have tiny values of their radii, which makes them hard to manufacture.

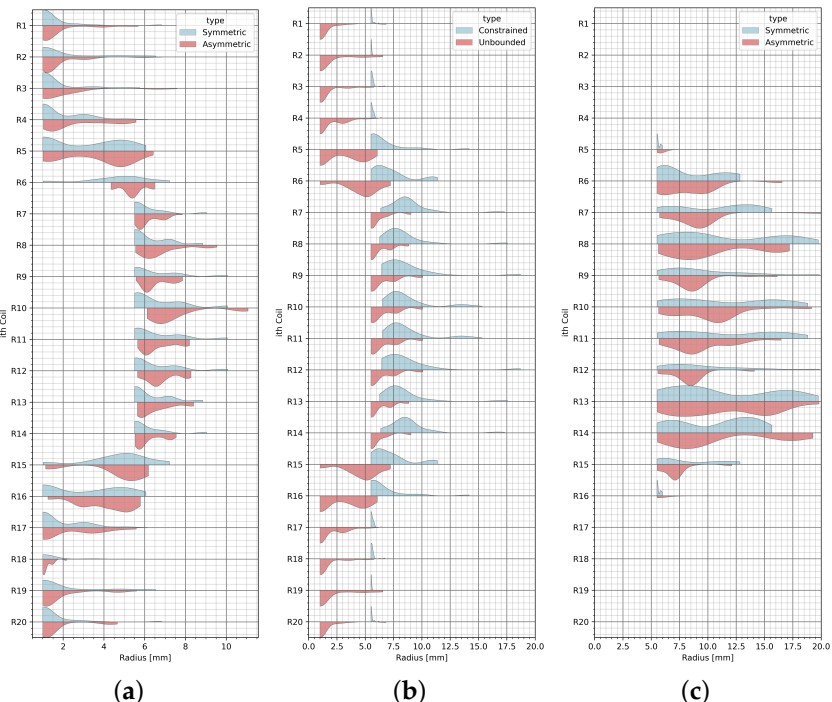

**Figure 10.** The distribution of the radii of the last 100 unsorted individuals. (**a**) Symmetric and asymmetric setups. All coils are allowed to move freely within the logical boundaries. (**b**) Only the symmetric setups where the coils are first allowed to move freely are shown (red); then, they are constrained in the 5.5–20 mm region (blue). (**c**) The distributions of the radii for the symmetric and asymmetric setups using only 12 coils. All coils were constrained to move within the 5.5–20 mm region.

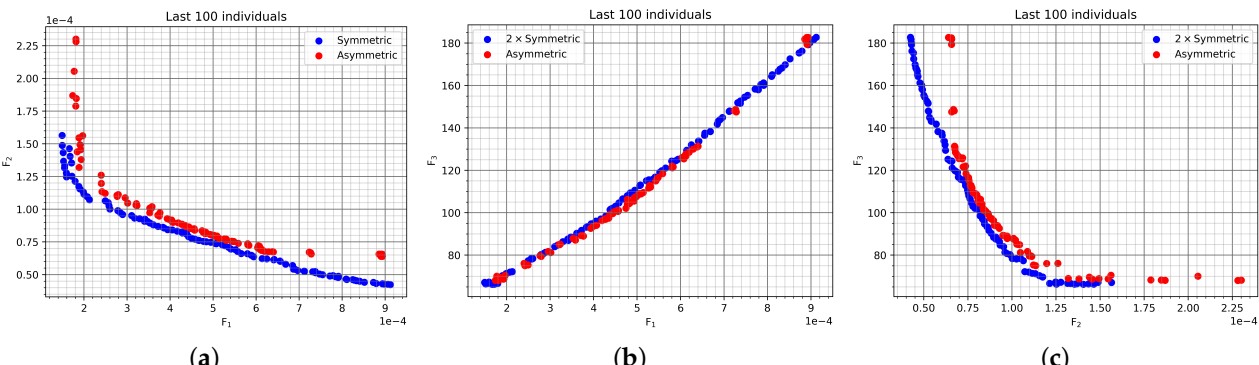

**Figure 11.** The last 100 individuals if the last generation is not sorted based on any of the conditions. These solutions are lying on the Pareto front. The blue dots show the symmetric setups, while the asymmetric solutions are depicted in red. (**a**) The distribution of individuals on the $F_1$–$F_2$ axis. (**b**) The distribution on the $F_1$–$F_3$ axis. (**c**) The distribution on the $F_2$–$F_3$ axis.

### 3.3. Optimization with the Parameter Settings of Cases (b) and (c)

The two other optimization runs were carried out on the symmetric variant to examine the impact on the Pareto surface if we neglected the last 3–3 turn from the optimization (case (c)) or did not allow radii smaller than 5.5 mm to be selected (case (b)).

First, in case (b), the optimization was performed with the constraint $5.5 \leq$ radii $\leq 50$ for all coils, as given in the original benchmark problem. This solution excludes the best candidates in the solution space, which have small radii. In the second experiment, we examined if the turns mentioned above were cut out, as described in case (c). The main question during this experiment is that of if we can resolve the task with only a twelve-turn coil instead of the original twenty-turn one.

The results of the optimization in case (b), where the minimum radii of the turns had to be greater than 5.5 mm, are plotted in Figures 10b and 12. Figure 10b shows the distribution of the optimal radii in the two examined cases. The red plot shows the distribution of the radii in the previous symmetric case, while the blue plot shows the new constrained solution. As can be seen from the picture, all of the radii are greater in this case. Therefore, the small turns at the two ends of the coil can significantly improve the uniformity of the magnetic field in the homogenized region. In Figure 12, we can see that the goal function values are significantly worse than in the previous case.

In case (c), we solved both the symmetric and the asymmetric problems with 12 coils instead of 20, as shown in Figure 12. The results can be seen in Figures 12 and 13. If the 100 cheapest solutions are considered, then the reduced number of coils produces better results, especially with an asymmetric setup. In terms of cost, contrary to what one would intuitively expect, the reduced setups cost more than their counterparts by 12–20%. The reason for this small difference is that the twelve-turn variant contains turns with generally bigger radii. This difference is not significant if we compare the price of the coil with the solution of the original problem (constrained, Figure 10b). Regarding the best $F_1$ and $F_2$ solutions, the reduced setups perform worse in every way than the 20-coil setups.

**Table 7.** Last individual in case (a).

|  | $F_1$ | $F_2$ | $F_3$ | $R_1$ | $R_2$ | $R_3$ | $R_4$ | $R_5$ | $R_6$ | $R_7$ | $R_8$ |
|---|---|---|---|---|---|---|---|---|---|---|---|
| Symmetric | $5.83 \times 10^{-4}$ | $3.22 \times 10^{-4}$ | $5.67 \times 10^1$ | 1.01 | 1.00 | 1.01 | 1.01 | 1.04 | 1.15 | 5.61 | 5.50 |
| Asymmetric | $5.41 \times 10^{-4}$ | $2.79 \times 10^{-4}$ | $6.86 \times 10^1$ | 1.16 | 1.12 | 1.01 | 1.67 | 1.15 | 4.35 | 5.84 | 6.07 |

|  | $R_9$ | $R_{10}$ | $R_{11}$ | $R_{12}$ | $R_{13}$ | $R_{14}$ | $R_{15}$ | $R_{16}$ | $R_{17}$ | $R_{18}$ | $R_{19}$ | $R_{20}$ |
|---|---|---|---|---|---|---|---|---|---|---|---|---|
| Symmetric | 5.50 | 5.54 | 5.54 | 5.50 | 5.50 | 5.61 | 1.15 | 1.04 | 1.01 | 1.01 | 1.00 | 1.01 |
| Asymmetric | 5.96 | 6.71 | 6.08 | 6.49 | 6.38 | 5.94 | 1.30 | 3.15 | 1.17 | 1.08 | 1.03 | 1.00 |

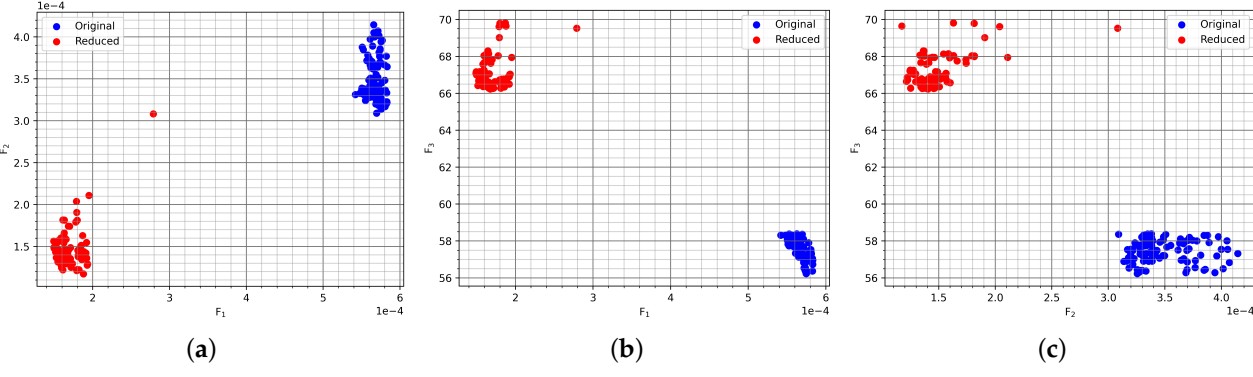

**(a)**　　　**(b)**　　　**(c)**

**Figure 12.** The comparison of the 100 cheapest symmetric solutions. With the blue color, all 20 coils are free to move, and with the red, only 12 coils are used, and they are constrained to move within the 5.5–20 mm region.

After analyzing the solution plot, the search range of the optimized turn radii can be reduced from 5.5–50 to 5.5–20 because the greater radii are not represented in the solution of the three objective problems.

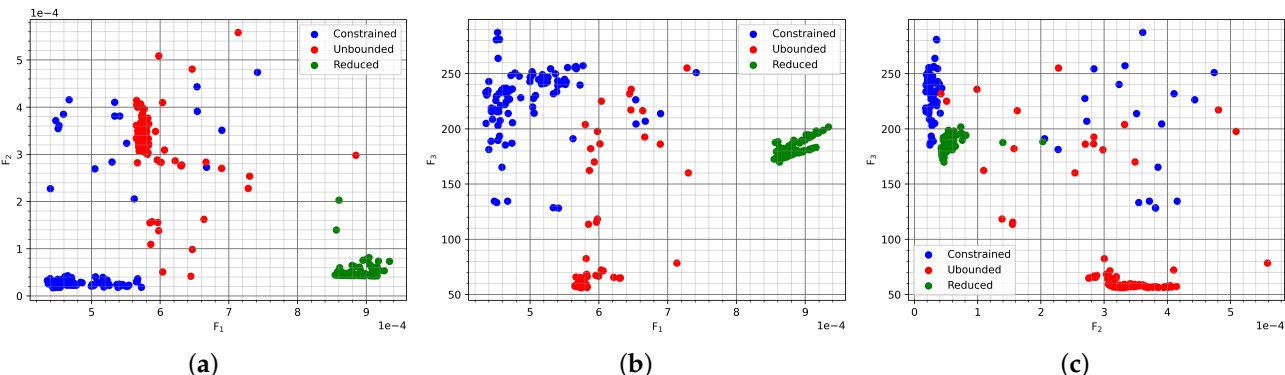

**Figure 13.** The image shows the last 100 symmetric solutions with various setups. The case where all 20 coils were free to move is in red. The blue color shows when all 20 coils are constrained in the 5.5–20 mm region, and finally, the green color depicts the case where only 12 coils are used, and they are constrained to the 5.5–20 mm region. (**a**) The distribution of the various setups on the $F_1$–$F_2$ axis, (**b**) on the $F_1$–$F_3$ axis, and (**c**) on the $F_2$–$F_3$ axis.

The proposed results and the working version of the realized optimization problem can be downloaded from the Ārtap project's homepage (the proposed solutions are available for download from: http://www.agros2d.org/artap/, accessed on 1 June 2021) together with a semi-analytical solution, and this can be used to validate the performance of FEM-based multi-objective optimization tools.

## 4. Conclusions

A novel three-variable analysis of the multi-objective TEAM benchmark problem is proposed in this paper. The original benchmark problem contains two similar two-variable Pareto optimizations. Firstly, the proposed coil is optimized to produce a uniform magnetic field in the examined region, and the sensitivity of the coil should also be minimized [12]. Secondly, the uniformity of the magnetic field and the mass of the coil are optimized. In practice, these two problems should be handled together. In this paper, the three objectives mentioned above—the uniformity ($F_1$), sensitivity ($F_2$), and price ($F_3$)—are considered simultaneously. Before optimizing the coil, an FEM simulation was made using two different tools with different mesh settings to find the right FEM setup. It was found that $F_1$ and $F_2$ are sensitive to the mesh settings, especially on the right side of the examined region. The FEMM-based calculation of the default (smart mesh function) mesh can produce 100% error in calculating $F_1$. It was found that the hp-adaptive solver was significantly faster and gave more accurate results. This solver was used to optimize with 1% in the error indicator, which produced an uncertainty of less than 20 μT in the results. This tolerance is acceptable because it is smaller than the measurement error or the Earth's magnetic field, which perturbs the results. This paper shows that this optimization problem is highly nonlinear, and nothing guarantees that only one optimal solution exists. We considered the asymmetrical cases in the solution and found that the symmetrical solutions always produced better solutions for $F_1$ and $F_2$. However, the difference between those solutions is insignificant for $F_1$ and $F_2$, as it is smaller than 20 μT.

Another interesting discovery is that the cheapest solutions are symmetrical setups, but they perform worse than the cheapest asymmetric ones for $F_1$ and $F_2$. We reduced the number of turns from 20 to 12, and we found that the price of the coil was reduced by only about 12–20%, which was below the expectations. Further studies should be carried out to validate the proposed results by performing measurements on at least a single layout, and these measurement results can be used to benchmark the proposed results.

**Author Contributions:** Conceptualization, T.O.; methodology, T.O., K.G.; software, K.G., T.O.; validation, K.G., T.O.; writing—original draft preparation, T.O., K.G.; writing—review and editing, T.O.; visualization, K.G., T.O.; supervision, T.O.; project administration, T.O.; funding acquisition, T.O. All authors have read and agreed to the published version of the manuscript.

**Funding:** This research received no external funding.

**Conflicts of Interest:** The authors declare no conflict of interest.

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
