# Peer review of "Robust and Multi-Objective Pareto Design of a Solenoid"

_electronics, doi:10.3390/electronics10172139_

Round 1

Reviewer 1 Report

The manuscript has English writing issues, whereas what the authors are trying to convey is not well represented by the words used. Let's take the abstract for example, the abstract is the most concise statement of the work, and should be the most well written part of a paper. Due to the huge number of writing problems, the reviewer could only list the abstract issues other than going to gory details of the main text.

  1. "Optimization is an essential part of the design of an electromagnetic device", define "optimization", what is optimized? One can even say "optimization is an essential part of everything", what is the intention of the statement?
  2. "Real world, multi-objective optimization problems are computationally expensive, and they often have parameters with unavoidable variations. " This reads strange by saying "...problems are computationally expensive". One needs to be specific about what is expressed. By switching words "real world, climate problems are computationally expensive, and they often have parameters with unavoidable variations", also says something, but not exactly. In other words, there is not much information in this sentence when readers are trying to refine your statement.
  3. "Therefore, the designers’ goal is not only to find a solution, which is feasible and its objectives are as good as possible, but these solutions should be insensitive to the parameter perturbations and manufacturing uncertainties." Wrong concept. Anything insensitive to the parameter perturbations and manufacturing uncertainties is called stationary. While stationary solution is not typically the goal of a design, which is better when phrased upon optimization problems.
  4. "This paper analyses a multi-objective TEAM benchmark problem, where the goal is to create a coil design, which can synthesize a uniform magnetic field distribution in the given region. The synthesized magnetic field should be uniform and robust for the manufacturing errors and as cheap as possible. This paper does not split this multi-objective optimization problem into two parts. It solves it as a three-objective optimization problem at once. Moreover, the paper investigates and shows the competitiveness of asymmetrical solutions in the case of a practical design." are you simply want to say that "we exampled with coil design in generating a uniform magnetic field as a multi-objective optimization problem" and the rest are details not tied to the abstract part?

In fact, even without the writing problem, it is almost impossible to validate the saying of the work for Solenoid, unless experimentally manufactured and matched in measurements. Otherwise, simulation demonstration must clearly state the modeling details, which is not presented in the work. Due to lack of details, the reviewer is not convinced that the work is valid.

Reviewer 2 Report

I find the work interesting and I congratulate the author. However, I think it is important to review some aspects of the document:
1.- It was very difficult for me to read the document due to the wording. This must be reviewed in depth in the document. There are paragraphs where the same word is constantly repeated. The text is redundant and makes reading difficult.
2.- It is convenient to review the wording of the abstract to clarify the proposal. In the abstract I did not understand the proposal correctly and this text is very relevant to the document. The author must clarify his proposal.
3.- The figures are very interesting. I like them. However, it is important to review the wording of your titles to clarify them and that the text is not redundant.
4.- I would like the proposed method to be explained more fully. With the information provided by the author, he would not be able to replicate the method. In that sense, I would also like to clarify the data used in the comparison.

Round 2

Reviewer 1 Report

The current writing appears to be reasonable for a journal paper.